**Investigation**

# The *Drosophila* tumor necrosis factor Eiger promotes Myc supercompetition independent of canonical Jun N-terminal kinase signaling

Albana L. Kodra,[1] Aditi Sharma Singh,[1] Claire de la Cova (iD),[2] Marcello Ziosi,[3] Laura A. Johnston (iD) [1,*]

[1]Department of Genetics and Development, Vagelos College of Physicians and Surgeons, Columbia University, New York, NY 10032, USA
[2]Department of Biological Sciences, University of Wisconsin, Milwaukee, WI 53201, USA
[3]New York Genome Center, New York, NY 10013, USA

*Corresponding author: Laura A. Johnston, Department of Genetics and Development, Vagelos College of Physicians and Surgeons, Columbia University, New York, NY 10032, USA. Email: lj180@columbia.edu

Numerous factors have been implicated in the cell–cell interactions that lead to elimination of cells via cell competition, a context-dependent process of cell selection in somatic tissues that is based on comparisons of cellular fitness. Here, we use a series of genetic tests in *Drosophila* to explore the relative contribution of the pleiotropic cytokine tumor necrosis factor α (TNFα) in Myc-mediated cell competition (also known as Myc supercompetition or Myc cell competition). We find that the sole *Drosophila* TNF, Eiger (Egr), its receptor Grindelwald (Grnd/TNF receptor), and the adaptor proteins Traf4 and Traf6 are required to eliminate wild-type "loser" cells during Myc cell competition. Although typically the interaction between Egr and Grnd leads to cell death by activating the intracellular Jun N-terminal kinase (JNK) stress signaling pathway, our experiments reveal that many components of canonical JNK signaling are dispensable for cell death in Myc cell competition, including the JNKKK Tak1, the JNKK Hemipterous and the JNK Basket. Our results suggest that Egr/Grnd signaling participates in Myc cell competition but functions in a role that is largely independent of the JNK signaling pathway.

Keywords: cell competition; Myc supercompetition; TNF; JNK; competitive signaling module; epithelium; *Drosophila*

## Introduction

Cell competition is an evolutionarily conserved process of somatic cell selection that has been implicated in numerous biological processes, including embryonic development, tissue maturation, organ size control (ctl) and cancer (Clavería *et al.* 2013; Sancho *et al.* 2013; Ellis *et al.* 2019; Sun *et al.* 2023; Tshering *et al.* 2023; Zhang *et al.* 2023). Cell competition leads to the elimination via death or expulsion of viable, but relatively weak cells ("losers") from growing tissues, allowing more robust cells ("winners") to populate the tissue (Johnston 2009). The specific mechanisms that recognize fitness differences and trigger elimination of the loser cells are influenced by the organism, tissue type, genetic background, and the particular circumstances of the competitive environment. First identified in *Drosophila*, cell competition employs context-dependent activation of signaling pathways involved in regulation of stress, immunity, or apoptosis. Signaling by the Jun N-terminal kinase (JNK) stress response pathway is often stimulated (Moreno, Basler, *et al.* 2002; de la Cova *et al.* 2004; Moreno and Basler 2004; Tyler *et al.* 2007; Igaki *et al.* 2009; Ohsawa *et al.* 2011; Kucinski *et al.* 2017; Banreti and Meier 2020). In some competitive contexts in *Drosophila*, components from the innate immune response are activated, forming a novel signaling module consisting of select components of the Toll and the immune deficiency (IMD) signaling pathways (hereafter called the CSM, for competitive signaling module) (Fig. 1a, left). The CSM

is best characterized for its mediation of the competitive interactions between wild-type (WT) cells and cells that express an extra copy of the conserved transcriptional regulator Myc (Meyer *et al.* 2014; Alpar *et al.* 2018; Nagata *et al.* 2019; Hof-Michel *et al.* 2024). In this work, we focus on interactions between the CSM and signaling from the TNF/TNF receptor (TNFR) pathway in the context of Myc-mediated cell competition.

Due to some shared components, the CSM resembles the innate immune system, but there are several notable differences. In contrast to IMD and Toll pathways, the CSM is not activated in immune tissues such as the fat body (FB) but is activated locally in mosaic wing discs in response to confrontation between competing cell populations (e.g. WT and Myc-expressing cells). Its activity is restricted to the competing cells and leads to the apoptotic elimination of the relatively less fit, "loser" cells (Alpar *et al.* 2018). Activation of the CSM results from transcriptional upregulation of the Spz proprotein and the serine proteases ModSP and Spz-processing enzyme (SPE) in the Myc supercompetitor cells. Production of these proteins leads to the proteolytic cleavage of Spz into an active ligand, allowing it to associate with one or more of 5 different Toll-related receptors (TRRs) that are highly expressed in the WT cells. This appears to restrict activation of the CSM to the WT cells, converting them into "losers" that induce Relish (Rel)/NF-κB-dependent expression of the Hid proapoptotic factor, and ultimately to their death (Fig. 1a) (Meyer *et al.* 2014; Alpar *et al.* 2018). The CSM thus consists of a novel configuration

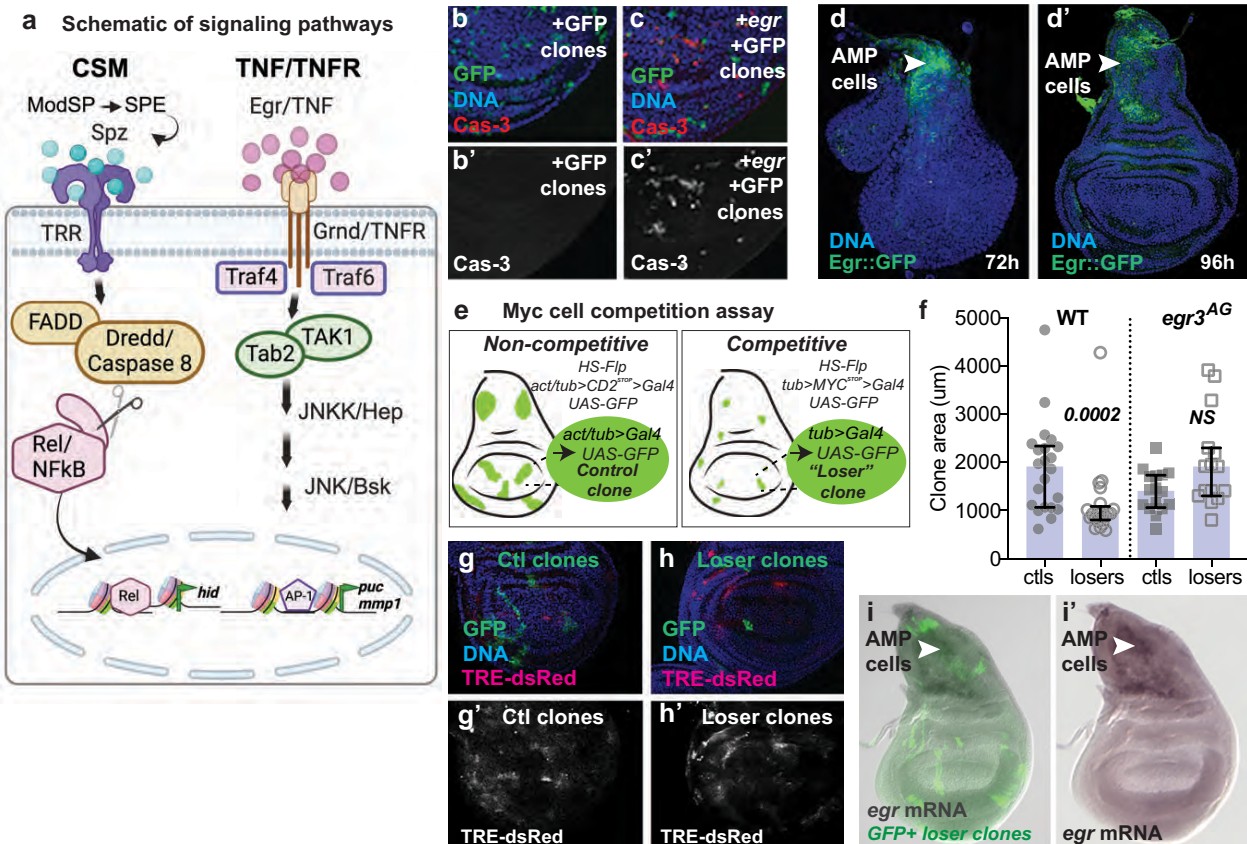

**Fig. 1.** Egr is required for the competitive elimination of WT loser cells in Myc-mediated cell competition. a). Schematic models of the CSM and TNF/TNFR signaling pathways. Left, the CSM pathway, based on data from (Meyer *et al.* 2014) and (Alpar *et al.* 2018), proposes that the Spz ligand, activated via processing by ModSP and SPE (Meyer *et al.* 2014) associates with the TRRs Toll 1 (Tl), possibly as a heterodimer with Toll-8 (Tollo). Toll-2 (18w), Toll-3 (Mst-prox), and Toll-9 are also genetically required for the elimination of loser cells in Myc cell competition, but their roles remain to be determined (Meyer *et al.* 2014; Alpar *et al.* 2018). Transduction of the Spz/Tl association requires the function of the TIR domain protein dSarm1/Ect4 (not shown), FADD, Caspase-8/Dredd, and Rel/NFkB to lead to loser cell death, and Rel requires cleavage by Dredd to induce apoptosis of the WT loser cells (Meyer *et al.* 2014). The direct targets of Rel in cell competition remain to be determined, but the *hid* gene is activated in a Rel-dependent manner. The antimicrobial peptide genes that Rel activates in innate immunity are not induced during cell competition (Meyer *et al.* 2014). Right, model of conventional TNF/TNFR signaling in *Drosophila* (Colombani and Andersen 2023). The TNF Egr associates with Grnd/TNFR on the plasma membrane of the cell. Traf4 and Traf6 function as adaptors within the cell and activate the core kinase cascade of the JNK pathway that includes Tak1/MAPKKK, Tab2, Hep/JNKK, and Bsk/JNK. Targets of the JNK pathway include the genes encoding the metalloprotease Mmp1 and the phosphatase Puckered, among others. b, b'). ctl, GFP-marked clones induced in wing discs and allowed to grow for 24 h. b') Anti-activated Cas-3 immunostaining. c, c'). Expression of *egr* in GFP-marked clones for 24 h leads to caspase activity (Cas-3) and elimination of cells from the wing disc. c') Anti-activated Cas-3 staining. d, d') Egr::GFP is highly expressed in the AMP cells under the disc notum (arrowhead) but is very low or undetectable elsewhere, in both early third instar (d, 72 h AEL) and mid-third instar wing discs (d", 96 h AEL). e) Schematic of Myc cell competition assay. Left, Flp-FRT recombination is used to generate wing disc cell clones marked by expression of UAS-GFP that are allowed to grow for defined periods of time. Measurement of the area of these ctl clones yields information on clonal growth in a noncompetitive context. Right, in parallel experiments with all cells expressing the *tub > Myc$^{STOP}$>* cassette (3xMyc cells), a competitive context is generated in the wing disc with the flp-out induction of GFP-marked clones (2xMyc). These "loser" clones compete with surrounding 3xMyc cells for occupancy of the wing disc. Competition is assessed via comparison of wing disc clone size in each context in WT and/or mutant genetic backgrounds. See "Materials and methods" for details. f) Graph of clone area measurements from competition assays in wing discs from WT and *egr3$^{AG}$* mutant backgrounds. In a WT background (left), loser clones grow significantly less than the noncompetitive ctl clones due to their competitive elimination (de la Cova *et al.* 2004). In the null *egr3$^{AG}$* mutant background (right), loser clones grow as well as their cognate ctl clones, indicating suppression of the loser phenotype. Clones grew for 50 ± 2 h. Error bars show median and interquartile range. Statistical significance is shown for comparisons of losers vs ctls and was determined by non-parametric Mann–Whitney tests. g, h) The activity of the TRE-dsRed JNK reporter is not specifically associated with either ctl or loser clones. ctl g) and loser h) clones were generated in the background of TRE-dsRed. TRE activity is apparent in both contexts but does not correlate with either ctl or loser status. i) RNA in situ hybridization to examine *egr* mRNA expression in loser clones in tub > *myc$^{STOP}$* > Gal4 wing discs. Loser clones are marked by GFP expression (green in i), overlaid on the *egr* mRNA (shown alone in i'). *egr* mRNA is easily detected in the AMP cells (arrowheads) but is not detectable in any of the loser clones nor in the surrounding tub > *myc$^{STOP}$*-expressing cells. NS, not significant (note: in all graphs, clone area was measured in microns). Clones labeled ctls were produced by FLP-out of the *act > CD2$^{STOP}$ > Gal4* (or *tub > CD2$^{STOP}$ > Gal4*, noted as *act/tub > CD2$^{STOP}$ >*) cassette and clones labeled losers were produced by FLP-out of the *tub > myc$^{STOP}$ > Gal4* cassette).

of immune response components that activate a genetic program distinct from that of the conventional *Drosophila* immune response (Meyer *et al.* 2014; Alpar *et al.* 2018).

Reporters of the JNK stress pathway such as *puc-lacZ*, an enhancer trap insertion in the *puckered* (*puc*) locus (encoding a phosphatase antagonist of JNK activity) (Martin-Blanco *et al.* 1998; McEwen and Peifer 2005), indicate that JNK activity is present in some epithelial cells of the wing and eye imaginal discs during their normal growth, where Puckered promotes cell survival by blocking JNK activity (McEwen and Peifer 2005; Willsey *et al.* 2016). JNK activity has also been observed in wing discs during Myc supercompetition, where *puc-lacZ* can be variably activated in either loser or

winner cells (de la Cova *et al.* 2004; Moreno and Basler 2004). However, the importance of JNK activity in loser cell elimination in Myc cell competition has remained unclear. In one report, only 30% of loser cell death was suppressed by the absence of JNK signaling (de la Cova *et al.* 2004). In another, suppression of loser cell death by blocking JNK signaling also required expression of p35, a pan-caspase inhibitor (Moreno and Basler 2004). By contrast, the death of WT loser cells in Myc supercompetition is completely suppressed by the genetic loss of any individual component of the CSM (Fig. 1a, left) (Meyer *et al.* 2014).

Cell death mediated by JNK signaling is induced via the highly conserved and pleiotropic TNFα signaling pathway in mammals and in *Drosophila* (Nagata 1997; Igaki and Miura 2014; Colombani and Andersen 2023). The sole *Drosophila* TNF, known as Eiger (Egr), has homology with mammalian ectodysplasia-A2 and other TNF ligand superfamily proteins (e.g. RANKL, CD40L, FasL, TNFα, and TRAIL) (Igaki *et al.* 2002). Like other TNFs, Egr is a type II transmembrane protein that also can be cleaved and glycosylated, allowing it to be secreted (Kauppila *et al.* 2003). Egr binds to the TNFRs Wengen (Wgn) (Kanda *et al.* 2002; Kauppila *et al.* 2003) and Grindelwald (Grnd) (Andersen *et al.* 2015). The Egr/TNFR complex, via the Traf 4 and Traf 6 adaptors, activates the core kinase cascade of the JNK pathway (Fig. 1a, right), which includes the MAP3K/JNKK kinase Tak1, the Tak1-associated binding protein 2 (Tab2), the MAP2K/JNK kinase Hemipterous (Hep), and the MAPK/JNK Basket (Bsk) (Kanda *et al.* 2002; Kauppila *et al.* 2003; Andersen *et al.* 2015). Egr/TNFR signaling is activated by numerous kinds of stress, including the innate immune response, where it cooperates with the IMD pathway to activate Rel-mediated expression of antimicrobial peptides in the FB, an adipose tissue and major immune organ in *Drosophila* larvae (Delaney *et al.* 2006). Under these conditions, Tak1 is a critical mediator of Rel activation, which requires both proteolytic cleavage by the caspase-8 homolog, Dredd, and phosphorylation by the IkB kinase (IKK) complex (Stoven *et al.* 2003; Erturk-Hasdemir *et al.* 2009). Tak1 is thus a key signaling component shared by the IMD/Rel pathway and the core JNK cascade triggered by Egr/TNFR (Kleino *et al.* 2005; Delaney *et al.* 2006). Interactions between JNK signaling and the Toll immune pathway have also been described (Wu *et al.* 2015).

With the goal of shedding light on the relative contribution of the JNK signaling pathway in Myc-induced cell competition, in this work, we used genetic tests to understand its role in the death and elimination of the WT "loser" cells. Since Egr activates cell death through JNK signaling (Andersen *et al.* 2015), we investigated the role of Egr and its receptors Grnd and Wgn in the activation of JNK signaling during Myc cell competition. We show here that genetic loss of Egr or its receptor Grnd, but not loss of the Wgn/TNFR, robustly blocks the death and elimination of loser cells in competition with cells that express more Myc. Genetic experiments using cell competition assays suggest that Egr/Grnd activity in the WT loser cells requires the function of Traf4 and Traf6. However, our results reveal that downstream of these adaptors, canonical JNK signaling is not necessary for loser cell elimination in this competitive context: genetic experiments indicate that Tak1 and the downstream JNK kinase Hep are dispensable, and a dominant negative form of the kinase Bsk, although relieving some of the death, is unable to fully suppress the competitive elimination of the WT loser cells. Thus, although Egr/Grnd function is required in the WT loser cells for their death, the activity they provide in this context is largely independent of JNK signaling. Instead, our results suggest that Egr/Grnd augments the CSM in the killing of WT loser cells. We propose that Egr/Grnd signaling participates in Myc supercompetition primarily by promoting the activity of the CSM.

## Materials and methods

### Fly strains and husbandry

Flies were raised at 25°C on standard cornmeal–molasses food (R-food, LabExpress) supplemented with fresh dry yeast. The following strains were used: $ywhsFlp^{122}$, $hep^{r75}$, $traf4^{ex1}$, $wgn^{Pe00637}$, $grnd^{M105292}$, $grnd^{DfBSC149}$, $Tak1^1$, $Tak1^2$, $UAS$-$Bsk^{DN}$, and $rnGal4$ (from Bloomington Drosophila Stock Center [BDSC]); $TRE$-$dsRed$ $16$ (gift of D. Bohmann); $wgn^{KO}$ and $Egr$::$GFP$ (fTRG) (gift of M. Milan); $egr^{3AG}$ (Kodra *et al.* 2020); $UAS$-$Grnd$-$IR^{KK}$ (Vienna Drosophila Resource Center); $UAS$-$egr^{IR}$ and $UAS$-$dTRAF4^{IR}$ (gifts of M. Miura); $UAS$-$egr^{#5}$/ $CyO$ and $UAS$-$dTRAF6^{IR}$ (gifts of T. Igaki); $UAS$-$grnd^{intra}$, encoding a dominant active form of Grnd, and $UAS$-$grnd^{extra}$ (gifts of P. Leopold); $act > CD2^{STOP} > Gal4$ (Neufeld *et al.* 1998); $tub > CD2^{STOP} > Gal4$ (Moreno and Basler 2004); and $tub > myc^{STOP} > Gal4$ (de la Cova *et al.* 2004).

### Cell competition assays

Competitive and noncompetitive ctl clone assays were performed in WT, homozygous mutant, or hemizygous mutant backgrounds as indicated (de la Cova *et al.* 2004, 2014; Meyer *et al.* 2014; Alpar *et al.* 2018). Eggs from appropriate crosses were collected on freshly yeasted grape plates for 2–4 h (or as indicated in figures) and allowed to develop at 25°C in a humid chamber for 24 h. After hatching, larvae were transferred to food vials supplemented with fresh yeast paste at densities of less than 40 larvae per vial to prevent crowding. To measure clonal growth and cell competition in a manner that allows precise temporal ctl of transgene expression, we used FLP recombinase-mediated, intramolecular recombination (reviewed in Germani *et al.* 2018). To generate competitive clones, a $tub > myc^{STOP} > Gal4$ cassette (where > represents a FLP-recognition target [FRT] site) was used to generate random GFP-marked $tub > Gal4$ clones (de la Cova *et al.* 2004). Prior to recombination, all cells express the *Myc* cDNA under ctl of the *tubulin* promoter, at a level less than 2-fold over endogenous *myc* levels (3xMyc) (Wu and Johnston 2010). Recognition of the FRTs by FLP leads to excision of the *myc* cDNA and transcriptional STOP sequence and generates cells that heritably express Gal4 ($tub > Gal4$) and can regulate expression of UAS genes of interest (e.g. GFP). As the *GFP*-expressing cells no longer express extra *myc*, they are subject to competition from surrounding 3xMyc cells that retained the $>myc^{STOP}>$ cassette (de la Cova *et al.* 2004) (Fig. 1e). FLP, under heat shock (HS) ctl, was activated in larvae carrying this cassette by HS at 37°C for 10 min, at 48 h after egg laying (AEL). Post-HS, larvae were allowed to grow at 25°C for 24, 48, or 96 h, as indicated for each experiment. In parallel, to randomly generate GFP-marked clones as ctls for noncompetitive clonal growth, larvae carrying the transgenic cassette, $act > CD2^{STOP} > Gal4$ (on chromosome 3) or $tub > CD2^{STOP} > Gal4$ (on chromosome 2), were used. The ctl clones were induced with a HS at 37°C for 6 min with the same timing as for the competitive Myc cassette (de la Cova *et al.* 2004, 2014; Meyer *et al.* 2014; Alpar *et al.* 2018). These HS times were optimized for each cassette to generate only few clones per disc, to avoid merged clones. A detailed protocol is available upon request.

### Tissue dissection, fixation, and imaging

Larval carcasses were dissected and inverted and fixed in 4% paraformaldehyde in phosphate-buffered saline (PF-PBS) for 20 min at room temperature, followed by several washes with 0.1% Tween-20 in PBS (PBTw). Hoechst 33258 (Sigma) or DAPI

(Invitrogen) was used to stain DNA. Wing discs were mounted in VectaShield Antifade on glass slides. Images were acquired with a Zeiss Axiophot with Apotome, Zeiss LSM800, or Leica LSM710 confocal microscope. Clone area (in microns) was measured with FIJI (ImageJ) software. Clones were scored in the central area of the wing disc (wing pouch [WP] and proximal hinge), where competition is most severe (Alpar *et al.* 2018). ctl GFP+ clones grown in parallel and subjected to the same experimental conditions (temperature and time of clone induction) were analyzed in parallel. At least 10 wing discs per genotype were scored. Maximum intensity Z-projection was performed on the stacks. Fiji (National Institute of Health [NIH], Bethesda, MD, USA) was used to measure the size of GFP-marked clones or the whole wing disc (based on DAPI staining). Nonparametric Mann–Whitney and Kruskal–Wallis tests were used to determine statistical significance as indicated in each figure.

## Immunohistochemistry

Primary antibodies used: guinea pig anti-Grnd (1:200, gift of P. Leopold), rabbit anti-Caspase-3 (Cas-3) (1:100, Cell Signaling), rabbit anti-GFP (1:2,000, Thermo Fisher), and rabbit anti-Dcp-1 (1:100, Cell Signaling) were incubated at 4°C overnight with dissected wing discs previously permeabilized with 0.5% Triton X-100 in PBS (PBTx) for 1 h and then blocked in PBTw with 5% normal serum for 1 h at room temperature (RT). The samples were incubated with the secondary antibodies Alexa555 and Alexa488 (1:600, Molecular Probes) (preabsorbed against fixed embryos) for 3 h at RT in the dark and then washed with PBTw either 3× at RT or overnight at 4°C. Hoechst 33258 or DAPI (Sigma) was used to stain DNA. Images were processed and where applicable, assembled from Z-stacks as maximum projections, with FIJI/ImageJ software.

## RNA in situ hybridization of wing discs

Dissected, fixed, and washed larval carcasses were added to 500 μl of 300 mM ammonium acetate and 500 μl of 100% ethanol and gently mixed, washed in 100% ethanol for 10 min at room temperature, washed in xylene/ethanol (1:1) for 10 min, washed in ethanol 3 times, 10 min each, washed in methanol for 2 min, washed in methanol/4% PF (1:1) for 2 min, fixed again in 4% PF-PBTw for 10 min, washed in PBTw 5 times for 5 min each, and incubated at RT in hybridization solution/PBTw (1:1) for 15 min; this was replaced by hybridization solution and incubated at the appropriate hybridization temperature for at least 1 h. Digoxigenin-RNA probes were added to hybridization solution, mixed well, and denatured at 90°C for 5 min followed by incubation on ice for 5 min. The hybridization solution was removed from the carcasses and replaced with the denatured probe in hybridization solution and incubated at the appropriate temperature overnight. Samples were washed and processed with anti-Digoxigenin antibodies (Roche) as in Alpar *et al.* (2018) A detailed protocol is available upon request.

## Reagent table

A list of reagents is included in Supplementary Table 1.

## Results

### The TNF Egr contributes to the elimination of WT loser cells

Reporters of JNK signaling activity are weakly activated in normally growing WT wing discs (McEwen and Peifer 2005; Harris *et al.* 2016) and have also been observed in a variety of conditions that lead to cellular stress. Indeed, JNK activity is important in a variety of contexts of cell competition (Igaki *et al.* 2009; Ohsawa *et al.* 2011; Igaki and Miura 2014; Kucinski *et al.* 2017). JNK activity has also been observed in Myc supercompetition, although in this context, the extent of its contribution to the loser cell fate has remained unclear (de la Cova *et al.* 2004; Moreno and Basler 2004). The core signaling components of the JNK pathway (Fig. 1a, right)—Traf4, Traf6, Tak1, Tab2, Hep, and Bsk—are all expressed in wing disc cells (modENCODE). Engagement of Egr with either of its receptors, Wgn and Grnd, allows Egr to be a powerful, JNK-dependent inducer of cell death in the developing eye and wing (Igaki *et al.* 2002; Moreno, Yan, *et al.* 2002; Andersen *et al.* 2015). This is illustrated by the expression of *egr* in GFP-marked cell clones in wing discs, which leads to the death and elimination of most of the GFP-positive cells in the WP within 24 h (Fig. 1b and c). Many dying cells, visualized using an antibody specific to cleaved and activated Cas-3, are visible in a basal focal plane of the wing disc (Fig. 1c'). To assess the role of Egr in Myc-mediated cell competition, we examined its endogenous expression in the wing disc using a reporter strain carrying an *egr::GFP* fusion protein (Sanchez *et al.* 2019). The growing wing disc epithelium consists of a monolayer of columnar cells (called the disc proper) that is contiguous to and covered by a squamous cell layer called the peripodium (Tripathi and Irvine 2022). The wing disc also houses the tracheal primordial cells and the adult myoblast precursor (AMP) cells that lie beneath the 2 epithelial layers in the notum (future adult dorsal thorax) (Tripathi and Irvine 2022). During the disc's rapid growth period in the second to mid-third larval instars (L2-mid-L3), Egr is highly expressed in the nonepithelial AMP cells and some tracheal cells (Fig. 1d and d') (Brown *et al.* 2014; Casas-Vila *et al.* 2017; Everetts *et al.* 2021). In contrast, in the WP epithelium, which gives rise to the adult wing blade, its expression is seemingly random and limited to a few cells primarily in the "transition zone," cuboidal cells that lie at the interface between the peripodial and columnar epithelial layers (Fig. 1d and d' and data not shown) (McClure and Schubiger 2005). Egr is also expressed in many other tissues including the FB, from which its soluble form is secreted into the circulating hemolymph and accessible to wing discs and other larval organs (Agrawal *et al.* 2016; de Vreede *et al.* 2022).

To investigate the role of Egr in the activation of JNK signaling in wing discs during Myc cell competition, we examined the expression of the JNK-specific *TRE-dsRed* reporter, which consists of multimerized AP-1-binding sites under ctl of a basal promoter (Chatterjee and Bohmann 2012), in the context of the Myc cell competition assay (Fig. 1e, g, and h). The basis of this cell competition assay is a transgenic FRT cassette, *tub > myc^{STOP} > Gal4*, that when intact, drives *myc* expression less than 2-fold above its endogenous level (cells have 3xMyc; see "Materials and methods" for details; Fig. 1e) (Wu and Johnston 2010). Flp-mediated excision of the >*myc^{STOP}* sequences allows heritable expression of Gal4 (*tub > gal4*) and activation of UAS-GFP and forms clones that express only endogenous Myc (i.e. 2xMyc), which are outcompeted by the surrounding nonclonal, 3xMyc cells (Fig. 1e, competitive context). CSM-mediated signaling between the 3xMyc "winner" cells and the 2xMyc "loser" cells leads to the induction of the proapoptotic gene *hid* in the loser cells (de la Cova *et al.* 2004, 2014; Meyer *et al.* 2014). Using this assay to induce HS-generated GFP-marked clones, we observed weak, sporadic activation of *TRE-dsRed* in wing discs in both noncompetitive ctl and competitive contexts, with no direct correlation of TRE activity in either ctl or loser clones (Fig. 1g and h). These results indicate that JNK activity is present under these conditions at low levels in the wing disc

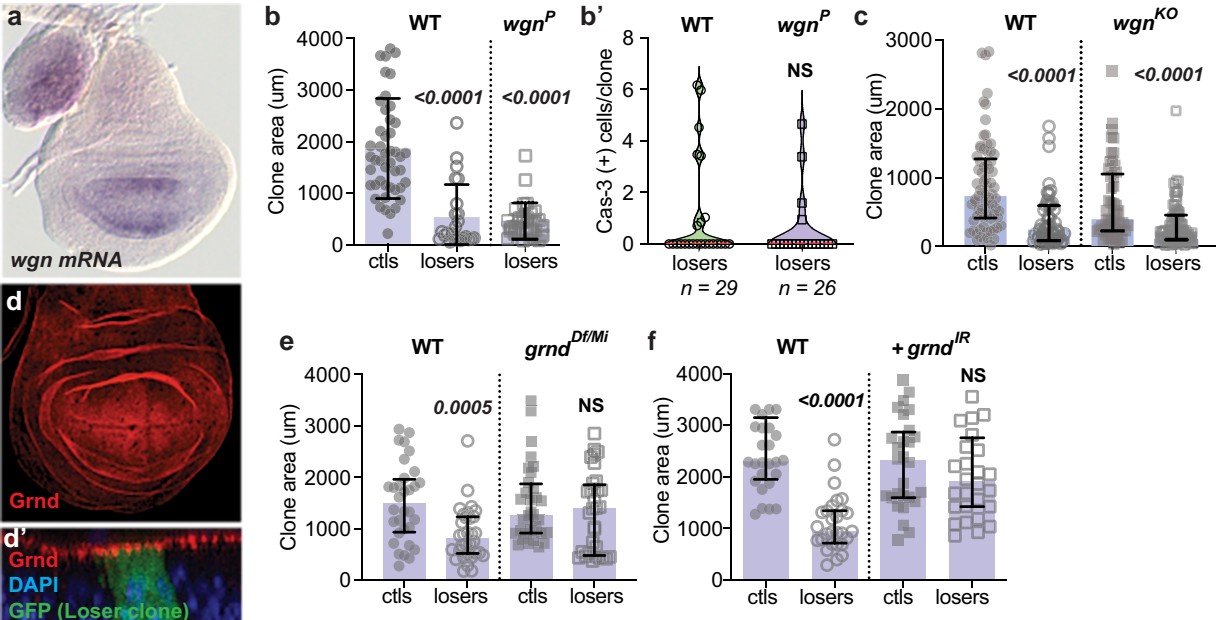

**Fig. 2.** The TNFR Grnd, but not the TNFR Wgn, contributes to the elimination of loser cells. a) *wgn* mRNA is expressed primarily in the WP region of the mid-L3 wing disc (disc is approximately 90 h AEL). b) *wgn* loss does not suppress loser cell elimination. Loser clones in both WT male and in hemizygous *wgn*$^P$ male larvae grow to smaller sizes compared to WT ctl clones. b') Loss of Wgn does not prevent cell death of the losers. Violin plots of active Cas-3-positive cells measured 24 h ACI in WT and *wgn*$^P$ mutants. Wing discs from male larvae were scored. $n$ = number of clones analyzed per genotype. c) Competition assay comparing ctl and loser clones in a WT background with those generated in parallel in *wgn*$^{KO}$ null mutants. Cell clones were measured in wing discs from male larvae (*WT/Y* and *wgn*$^{KO}$*/Y*). d, d') Grnd protein is uniformly expressed and apically localized in wing disc cells (disc is 120 h AEL; a maximum projection of a z-series is shown). d') Cross-section showing apical Grnd localization in wing disc with GFP-marked WT loser clone (in the *tub > myc*$^{STOP}$ *> Gal4* background). e, f) Loss of *grnd* suppresses the elimination of loser cells under competitive conditions but does not alter ctl clone size in the noncompetitive context. e) competition assays done in the background of the *grnd* null mutants *grnd*$^{Df}$/*grnd*$^{Mi}$ in *trans*. f) Competition assays in which *grnd*-RNAi (*grndIR*) was expressed specifically in the clones. Clones in b, c) and e, f) grew for 48 ± 2 h. Error bars represent median and interquartile range. Statistical significance is shown for comparisons of losers vs ctls and was determined by nonparametric Mann–Whitney and Kruskal–Wallis tests. NS, not significant (note: in all graphs, clone area was measured in microns. Clones labeled ctls were produced by FLP-out of the *act > CD2*$^{STOP}$ *> Gal4* (or *tub > CD2*$^{STOP}$ *> Gal4,* noted as *act/tub > CD2*$^{STOP}$*>*) cassette and clones labeled losers were produced by FLP-out of the *tub > myc*$^{STOP}$ *> Gal4* cassette).

epithelium but shows no overt connection to a cell's specific competitive status.

To assess whether Egr contributed to the death of the loser cells, we tested whether loss of Egr, using the *egr*$^{3AG}$ null allele (Kodra *et al.* 2020), altered the fate of loser cells in the competition assay. In a WT background of cells expressing *tub > myc*$^{STOP}$ *> Gal4*, excision of the *>myc*$^{STOP}$ cassette led to the generation of GFP-marked loser clones that were competitively eliminated from the wing disc, leading to a significant reduction in clone size compared to noncompetitive ctl clones generated in parallel (Fig. 1f, WT ctls and losers). However, in the *egr*$^{3AG}$ mutant background, clones generated in both competitive and noncompetitive contexts grew at the same rate and to a similar final size (Fig. 1f, *egr*$^{3AG}$ ctls and losers), indicating that in the absence of *egr*, competition between the *tub > myc*$^{STOP}$*>*-expressing cells and the clones of WT cells was abolished. Contrary to expectations, we were unable to detect *egr* transcripts within the loser cell clones in competition assays by RNA in situ hybridization (Fig. 1i), suggesting that Egr was produced elsewhere in the wing disc (e.g. AMP cells) or from another tissue in the larva (data not shown; Sharma Singh, in prep). These results indicate that Egr is genetically required for the elimination of loser cells in wing discs in the *tub > myc*$^{STOP}$ *> Gal4* background but possibly functions in this context in a paracrine manner.

## Grnd/TNFR is required to eliminate WT loser cells from wing discs

The *Drosophila* genome encodes 2 TNFRs, Wgn and Grnd, both of which are abundantly expressed in wing discs. Egr binds to both

Wgn and Grnd and can activate downstream signaling that requires Traf4 and Traf6 through each receptor (Igaki *et al.* 2002; Kanda *et al.* 2002; Geuking *et al.* 2005; Andersen *et al.* 2015). Previous reports of *egr* overexpression in the eye disc linked Egr activity to Wgn, the adaptors Traf4 and Traf6, and cell death via the JNK signaling pathway (Igaki *et al.* 2002; Kanda *et al.* 2002; Geuking *et al.* 2005). Although Egr expression in the wing disc is largely restricted to the nonepithelial AMP cells, *wgn* mRNA is robustly expressed in the WP (Fig. 2a), the region of the disc subject to the most severe effects of cell competition (Alpar *et al.* 2018). As *wgn* is located on the X chromosome, we examined clone growth and cell competition in wing discs from WT and *wgn*$^P$ mutant male larvae. As expected, after a 48-h growth period, loser cells in WT wing discs were outcompeted by *tub > myc*$^{STOP}$ *> Gal4* cells, yielding clones that were significantly smaller than noncompetitive ctl clones (Fig. 2b, WT ctls vs WT losers). Loser clones in wing discs from *wgn*$^P$ mutants were just as efficiently outcompeted as in WT wing discs (Fig. 2b), with similar numbers of dying cells per clone (Fig. 2b'). As the *wgn*$^P$ allele is likely hypomorphic, we also tested *wgn*$^{KO}$, a targeted knockout allele (Andersen *et al.* 2015). Similarly, loser cells were efficiently eliminated in the competition assay when *wgn* was completely lacking (Fig. 2c). These results suggest that Wgn is not required for the elimination of WT loser cells and are consistent with recent work reporting that Wgn has limited function in Egr signaling in wing imaginal discs (Palmerini *et al.* 2021).

In contrast to Wgn, the TNFR Grnd is uniformly highly expressed in the wing disc epithelium and localized to subapical

epithelial cell membranes (Fig. 2d and d'). Although present at high levels, loss of Grnd does not affect cell viability under normal conditions (Andersen *et al.* 2015). When examined in cell competition assays, Grnd protein localized normally to the apical surface of cells in both loser and ctl clones of third instar larval wing discs (Fig. 2d'). To test whether Grnd functioned in cell competition, we used the *trans*-heterozygous combination $grnd^{Df}/grnd^{Minos}$, which expresses no detectable Grnd protein (Andersen *et al.* 2015). Strikingly, the absence of *grnd* suppressed most of the elimination of loser cells, allowing loser clones to grow to sizes similar to the ctls (Fig. 2e, $grnd^{Df/Mi}$ ctls vs losers). In contrast, *grnd* loss had no effect on the size of noncompetitive ctl clones (Fig. 2e, $grnd^{Df/Mi}$ ctls vs WT ctls), indicating that Grnd's role in loser clones was specific to the competitive context. To determine whether Grnd was required autonomously within the loser cells, we expressed *UAS-grnd-RNAi* ($grnd^{IR}$) in the loser cells. Again, knockdown of *grnd* impaired cell competition but had no effect on the growth of clones in a noncompetitive environment (Fig. 2f). Together, our results suggest that Egr functions with Grnd to promote the elimination of loser cells in wing discs. Because we found no evidence of Egr in the loser or winner cells themselves, we infer that Egr is produced elsewhere in the wing discs or in another larval tissue. We suggest that in this competitive context, Egr functions as a paracrine factor and uses Grnd as its receptor on wing disc cells.

Grnd binds Egr via its extracellular N-terminal Caspase Recruitment Domain (CRD) domain, which results in signaling transmitted through its intracellular C-terminal domain (Andersen *et al.* 2015). Expression of just the intracellular portion of Grnd, *UAS-grnd^{intra}*, which functions as a dominant activator of JNK signaling, leads to apoptosis in an Egr-independent manner (Andersen *et al.* 2015). Apoptosis induced by expression of *UAS-grnd^{intra}* is efficiently suppressed in the $hep^{r75}$ mutant background, a severe loss of function allele of the Hep/MKK7 protein that is required to transmit JNK signaling (Glise *et al.* 1995), confirming that Grnd functions upstream of the JNK signaling cascade (Andersen *et al.* 2015). To determine whether Grnd-mediated intracellular signaling is sufficient to eliminate loser cells in cell competition, we expressed *UAS-grnd^{intra}* in wing discs. As a ctl, we expressed *UAS-grnd^{extra}*, a transgene containing only the extracellular portion of Grnd, which binds to Egr but does not activate JNK signaling (Andersen *et al.* 2015). Expression of *UAS-grnd^{extra}* using *mGal4*, which drives expression in the WP, led to only a slight, patchy rise in cell death (Fig. 3a and a'), consistent with previous results (Andersen *et al.* 2015). *grnd^{extra}* also had no effect on growth when expressed in clones, in either a noncompetitive or a competitive environment (Fig. 3b and c). In contrast, *mGal4*-directed expression of *grnd^{intra}* led to massive cell death, substantially reducing the size of the wing discs (Fig. 3d and d'). Moreover, when we expressed *grnd^{intra}* in GFP-marked ctl or loser clones, despite being initially detectable (Fig. 3e and f), by 48 h after clone induction (ACI), wing discs from were completely devoid of GFP-positive cells, indicating that all of the clones were eliminated (Fig. 3g and h). Thus, our results confirm that expression of *grnd^{intra}*, and by inference Grnd activation, transmits a potent death-inducing signal within cells (Andersen *et al.* 2015). Importantly, complete elimination of *grnd^{intra}*-expressing clones from the wing epithelium was the outcome in both the noncompetitive (Fig. 3g) and the competitive environment (Fig. 3h).

## Traf4 and Traf6 contribute to elimination of WT loser cells

In *Drosophila*, intracellular signaling via TNFRs is transduced through the TNFR-associated factors Traf4 (formerly known as dTraf1) and Traf6 (formerly dTraf2). TNF–TNFR association has been shown to activate both the JNK and the NF-κB signaling pathways, possibly because different intracellular signals can be transduced by different Traf adaptors (Cha *et al.* 2003). Traf4 physically associates with Wgn and Grnd, as well as Tak1, Tab2, and the Sterile-20 kinase, Misshapen (Liu *et al.* 1999; Cha *et al.* 2003; Andersen *et al.* 2015). Expression of Traf4 is sufficient to induce JNK activity and cell death (Cha *et al.* 2003). Although signaling mediated by Traf4 is directed primarily through the JNK pathway, Traf4 has also been implicated in NF-κB-mediated responses (Shen *et al.* 2001; Ayyar *et al.* 2007), and Traf4 is required to activate the Rel NF-κB factor for maintenance of proneural gene expression in wing discs (Ayyar *et al.* 2007).

We tested Traf4's requirement in the elimination of WT loser cells by depleting its function in cell competition assays. Competition was suppressed when *UAS-Traf4^{IR}*, encoding a *Traf4* inverted repeat for RNAi-mediated interference (Igaki *et al.* 2006), was expressed specifically in the loser cells (Fig. 3i, losers), whereas its expression did not affect the growth of ctl clones (Fig. 3i, ctls). The elimination of loser cells was also suppressed in larvae carrying the severe *Traf4^{ex1}* allele (Cha *et al.* 2003) and resulted in loser clones that were comparable in size to ctl clones in a noncompetitive environment (Fig. 3j).

The Traf6 adaptor also associates with the intracellular domains of Wgn and Grnd (Andersen *et al.* 2015). Unlike Traf4, ectopic expression of Traf6 is not sufficient to activate JNK signaling nor induce apoptosis, but it can participate in both JNK and NF-κB signaling (Cha *et al.* 2003; Geuking *et al.* 2005). Traf6 functionally interacts with Pelle, the Toll pathway kinase, and in S2 cells, Pelle and Traf6 cooperate in the activation of the NF-κB dorsal (Shen *et al.* 2001). To examine the role of Traf6 in cell competition, we expressed the inverted repeat RNAi transgene, *UAS-Traf6^{IR}* (Igaki *et al.* 2006) in cell clones in both noncompetitive and competitive contexts. Like loss of *Traf4*, knockdown of *Traf6* in the loser cells suppressed their elimination, allowing the clones to grow to the same size as noncompetitive ctl clones (Fig. 3k). The loss of *Traf6*, like that of *Traf4*, had no obvious effect on cells in noncompetitive ctl clones, indicating that its role was specific to the loser cells in the competitive context. Thus, the specific elimination of loser cells requires the function of both Traf4 and Traf6, consistent with these adaptors acting with Egr/Grnd in the loser cells.

## JNK signaling is activated but has a minor role in elimination of WT loser cells

Ectopic expression of *egr* in a stripe bisecting the wing disc induces massive cell death and tissue distortion (Fig. 4a), which are both markedly suppressed in larvae mutant for $hep^{r75}$ (Fig. 4b). As this mutant removes Hep/MKK7 function (Glise *et al.* 1995), this result is a clear demonstration that Egr-induced cell death is mediated by the JNK signaling pathway. Previous work reported that in cell competition assays in wing discs, only a small fraction of loser cell death was suppressed by loss of *hep* (de la Cova *et al.* 2004). Consistent with this, induction of cell competition assays in a WT background caused 45% of wing disc loser clones to contain Cas-3-positive cells at 24 h ACI (Fig. 4c and e, WT). Similarly, in $hep^{r75}$ mutant wing discs at 24 h ACI, 42% of loser clones contained Cas-3-positive cells, although on average, they contained fewer Cas-3-positive cells/clone (Fig. 4d and e, $hep^{r75}$). This result confirms that loss of *hep* slightly reduces loser cell death but does not wholly prevent it. Moreover, $hep^{r75}$ loser cells were ultimately eliminated from the wing discs as effectively as WT loser cells: by 50 h ACI, the median size of loser clones in $hep^{r75}$ mutant males

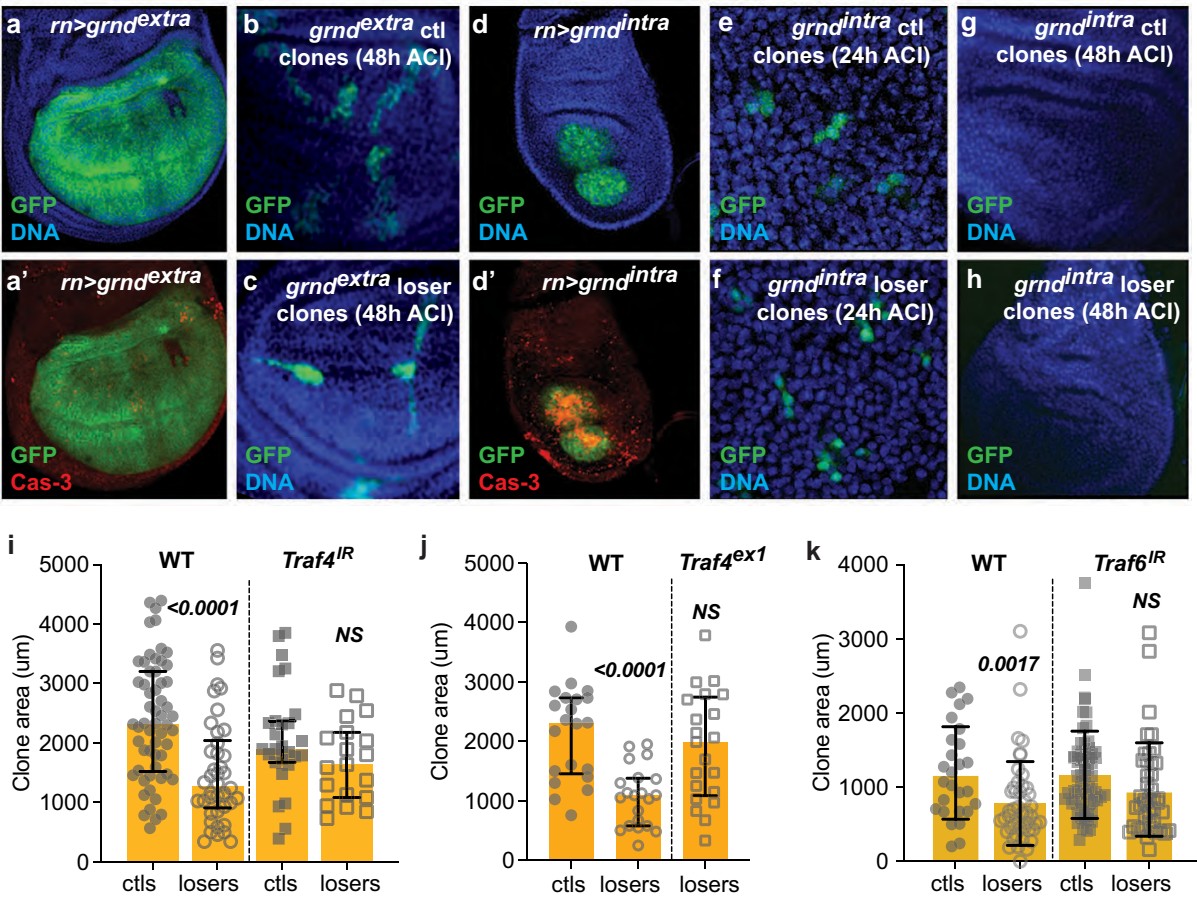

**Fig. 3.** Traf4 and Traf6 contribute to the elimination of loser cells. a) Expression of *grnd^extra* in the WP induces a small number of Cas3-positive cells a') but otherwise does not alter disc morphology. b) Representative wing disc showing noncompetitive ctl clones coexpressing *GFP* and *grnd^extra*. c) Representative wing disc showing loser clones in the competitive context coexpressing *GFP* and *grnd^extra*. d) Expression of a dominant active form of Grnd, *grnd^intra*, induces massive cell death d') and severely reduces overall wing disc size. e, f) Representative wing disc with *grnd^intra* expression in noncompetitive ctl clones e) and in loser clones f) in the competitive context, 24 h ACI. GFP-marked, *grndi^intra*-expressing clones are detectable in each context but are composed of only 2–3 cells. g, h) Wing discs with ctl g) or loser h) clones at 48 h ACI; no clones are detectable in either context, due to their elimination from the disc by *grnd*^intra expression during this time. i, j) While expression of *Traf4*-RNAi (*Traf4^IR*) in ctl clones does not alter their growth (i, WT), expression of *Traf4^IR* specifically in loser cells suppresses their elimination from the wing disc (i, *Traf4^IR*). j) Similarly, the elimination of loser cells is suppressed in the *Traf4^ex1* null mutant background. k) Loss of *Traf6* by expression of *Traf6*-RNAi (*Traf6^IR*) in the loser cells prevents their elimination. Clones in i, k grew from 48–96 h ± 2 h AEL. Error bars show median and interquartile range. Statistical significance is shown for comparisons of losers vs ctls and was determined by non-parametric Mann–Whitney tests. NS, not significant (note: in all graphs, clone area was measured in microns. Clones labeled ctls were produced by FLP-out of the *act > CD2^STOP > Gal4* (or *tub > CD2 ^STOP > Gal4*, noted as *act/tub > CD2^STOP>*) cassette and clones labeled losers were produced by FLP-out of the *tub > myc^STOP > Gal4* cassette).

was similar to that of loser clones in a WT background, and both were significantly smaller than noncompetitive ctl clones (Fig. 4f). In addition, expression in the loser clones of a dominant negative form of the Jun kinase, Bsk, which functions downstream of Hep (Riesgo-Escovar *et al.* 1996; Sluss *et al.* 1996), also resulted in clone sizes that were significantly smaller than noncompetitive ctls (Fig. 4g). However, these loser clones did grow slightly larger than WT loser clones, suggesting the dominant negative Bsk suppressed a fraction of the loser cell death.

The MAP3-K TGF-beta-activated kinase Tak1 is also critically required for transduction of JNK signaling downstream of Egr/Grnd activity (Andersen *et al.* 2015). However, in previous studies, Tak1 was found to be dispensable in the elimination of loser cells in Myc- and in $Rp^{-/+}$-mediated cell competition (Meyer *et al.* 2014). We reevaluated these results here using 2 different *Tak1* null alleles harboring point mutations that inactivate the kinase domain (Vidal *et al.* 2001; Delaney *et al.* 2006). We induced Myc cell competition in wing discs of male $Tak1^2/Y$ or $Tak1^1/Y$ mutant larvae and found that neither mutation suppressed the elimination of loser

cells (Fig. 4h), confirming that Tak1 does not contribute to the CSM (Meyer *et al.* 2014). Taken together, these experiments indicate that none of these essential signaling intermediates of the JNK pathway is sufficient to block the competitive loss of the loser cells from the wing disc.

## The function of Egr/Grnd in WT loser cells is not contingent upon JNK signaling

Our results establish that Egr activity, through its receptor Grnd, is required for the elimination of WT loser cells from the wing disc epithelium in Myc cell competition. However, our experiments also indicate that downstream of the Traf4 and Traf6 adaptors, canonical JNK signaling is either completely dispensable (Tak1) or only modestly contributes to loser elimination (Hep and Bsk), despite their typical activation downstream of Egr/Grnd signaling. These results suggest that in this competitive context, Egr/Grnd's role in loser cell elimination is largely independent of JNK activity. However, the absence of a requirement for Tak1 in the CSM raises an interesting conundrum, since on the one hand, Tak1 is

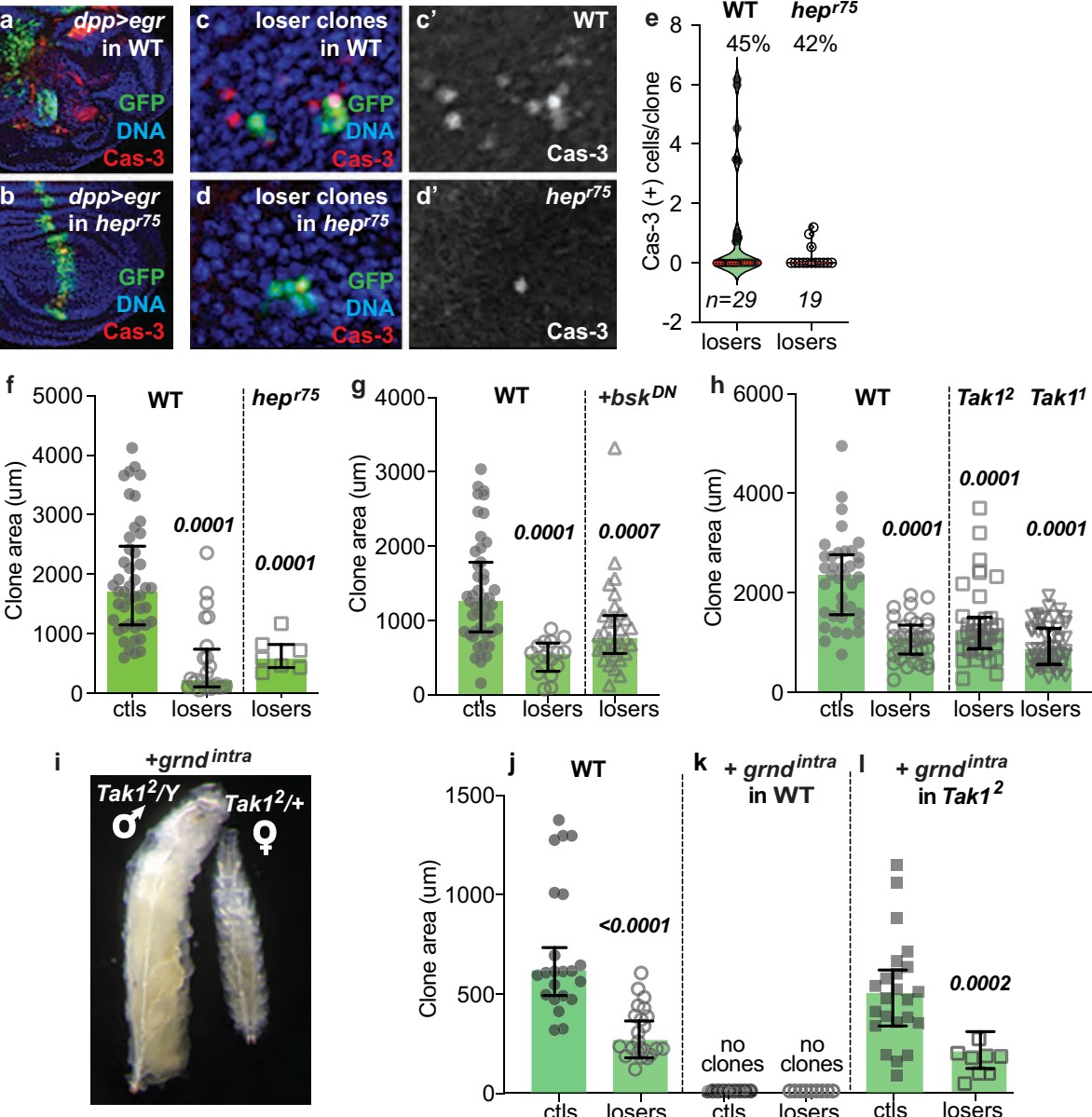

**Fig. 4.** Loss of JNK signaling downstream of the Traf adaptors has a minor effect on the elimination of loser cells in Myc-mediated cell competition. a, b) The *hep^r75* efficiently blocks cell death induced by ectopic Egr expression. Egr expression in a stripe bisecting the wing disc with *dppGal4* induces massive cell autonomous and noncell autonomous cell death and tissue distortion. *UAS-egr* was coexpressed with *UAS-GFP* in wing discs from WT a) or *hep^r75* male larvae b). The *hep^r75* allele efficiently blocks the apoptosis and tissue deformation induced by *UAS-egr*. Anti-activated Cas-3 (red) staining highlights dying cells. Wing discs were from male larvae (*WT/Y* or *hep^r75/Y*). c, c') In a WT background, GFP-marked loser clones die at high frequency, as shown with anti-activated Cas-3 staining c'). In the *hep^r75/Y* background, cell death in loser wing disc clones is reduced at 24 h ± 2 h ACI compared to loser clones in *WT/Y*, as shown by anti-Cas-3 staining. e). Violin plots with quantification of activated Cas-3 (+) cells/clone at 24 h ± 2 h ACI from the clones in c, d). The percent of loser clones containing Cas-3 (+) cells is shown at the top, and the number of Cas3 (+) cells per clone shown in the plots. *n* = number of clones scored per genotype (*WT/Y* and *hep^r75/Y*). f). By 50 h ACI, loser clones in a WT background are significantly smaller than noncompetitive ctls, and their competitive elimination is not suppressed in *hep^r75* mutants. Clones were scored in wing discs from male larvae (*WT/Y* and *hep^r75/Y*). g) Loser clones expressing *UAS-Bsk^DN* are significantly smaller than WT noncompetitive ctls, although larger than WT loser clones (P = 0.0040), suggesting some cell death was prevented. h) Loss of *Tak1* has no effect on the elimination of loser cells. Two different null alleles of Tak1 were tested (*Tak1^1* and *Tak1^2*). Clones were scored in wing discs from male larvae (*Tak1/Y* and *WT/Y*). i) Sibling *Tak1^2* mutant male (left) and female (right) larvae containing *UAS-grnd^intra*-expressing clones in the *tub > CD2^STOP > Gal4* context. The complete loss of *Tak1* in hemizygous *Tak1^2* males abolishes the massive death induction due to expression of *Grnd^intra* (e.g. Fig 3d) and allows normal larval growth, whereas female *Tak1^2/+* larvae remain susceptible to *Grnd^intra*-mediated cell killing and are thus severely reduced in size. j–l) The *Tak1^2* mutant suppresses *grnd^intra*-induced apoptosis in both ctl and loser clones, but it does not prevent cell competition (CSM)-induced apoptosis in the loser clones. j) ctl and loser clones from *WT* males. k) Neither ctl nor loser clones expressing *grnd^intra* grow, due to extensive cell death (see Fig. 3d); represented here as "no clones." l) Under noncompetitive conditions, *Tak1^2* suppressed the cell death induced by *grnd^intra* expression in ctl clones (+*grnd^intra* in *Tak1^2* ctls in l vs WT ctls in j, P = 0.3256). In contrast, in the competitive context, although the *Tak1* mutant suppressed the effect of *grnd^intra* in the cells and allowed them to form clones, the cells are still subject to cell competition via the CSM (+*grnd^intra* in *Tak1^2* losers vs ctls, P = 0.0002). In f) and g), clones grew for 48 ± 1.5 h; in h) and j–l), clones grew for 48 ± 3 h. Error bars in graphs show median and interquartile range. Statistical significance is shown for comparisons of losers vs ctls and was determined by nonparametric Mann–Whitney tests for each condition compared to ctls and Kruskal–Wallis (across genotypes) tests. NS, not significant (note: in all graphs, clone area was measured in microns. Clones labeled ctls were produced by FLP-out of the *act > CD2^STOP > Gal4* (or *tub > CD2^STOP > Gal4*, noted as *act/tub > CD2^STOP >*) cassette and clones labeled losers were produced by FLP-out of the *tub > myc^STOP > Gal4* cassette).

required downstream of TNFR to activate JNK signaling (Vidal *et al.* 2001; Delaney *et al.* 2006; Stronach *et al.* 2014; Andersen *et al.* 2015), and on the other, it plays crucial roles in the activation of Rel in the immune response to pathogenic infection (Silverman *et al.* 2003; Park *et al.* 2004).

Previous work demonstrated that as part of the CSM, the NF-κB factor Rel is required and sufficient for the expression of Hid and apoptosis of WT loser cells (Meyer *et al.* 2014; Nagata *et al.* 2019; Hof-Michel *et al.* 2024). In the same competitive context, we find here that the absence of either *egr* or *grnd* is also sufficient to suppress loss of loser cells. One way to explain these findings is if signaling from Egr/Grnd in the loser cells participates in or reinforces the CSM. If so, we reasoned that since Tak1 activity is not required for the CSM to mediate the elimination of loser cells (Fig. 4h) (Meyer *et al.* 2014), the *Tak1* mutant should not suppress Egr/Grnd activation in loser cell death. Conversely, in a noncompetitive context, loss of *Tak1* would be predicted to completely block the canonical JNK signaling downstream of Egr/Grnd activation and allow the cells to survive (Andersen *et al.* 2015).

We tested these ideas by clonally expressing the dominant-activating transgene, *grnd^intra^*, in *Tak1^2/Y^* hemizygous male larvae and quantifying the size of the clones in wing discs in both non-competitive ctl and competitive contexts. Female *Tak1^2/+^* larvae, which clonally expressed *grnd^intra^* but were heterozygous for *Tak1*, exhibited severely diminished larval growth due to the massive death induced by *grnd^intra^* and thus served as an internal ctl (Fig. 4i) (Andersen *et al.* 2015). By contrast, hemizygous *Tak1^2/Y^* males from the same cross were unable to transduce the *grnd^intra^*-mediated death signal, thus allowing the larvae to develop normally (Fig. 4i).

When cell competition was induced in WT male larvae, we observed the expected growth disadvantage of loser cells compared to noncompetitive ctls (Fig. 4j). In striking contrast, expression of *grnd^intra^* in either ctl or loser clones led to acute cell death (e.g. Fig. 3d–h) and completely blocked their growth (Fig. 4k, represented as "no clones"). As predicted, in the noncompetitive context, loss of *Tak1* function efficiently blocked *grnd^intra^*-induced cell death and allowed the clones to grow nearly as well as the WT ctls (ctls, Fig. 4l vs j, P = 0.3256). In the competitive *tub > myc^STOP^>* context, *Tak1* loss also allowed the growth of *grnd^intra^*-expressing clones (losers, v 4 l). However, these clones were still significantly smaller than their cognate noncompetitive ctl clones (Fig. 4l, ctls vs losers, P = 0.0002), again providing evidence that in the competitive context, loss of *Tak1* did not interfere with the acquisition of the "loser" fate in those cells (Meyer *et al.* 2014). These results demonstrate that while Tak1 depletion selectively prevented JNK-dependent cell death driven by *grnd^intra^*, the CSM-induced loser cell death still occurred.

Taken together, our experiments argue that in Myc-mediated cell competition, Egr/Grnd activity reinforces the CSM-mediated cell death response in the WT loser cells that disadvantages them and restricts loser clone growth. However, they also argue that Egr/Grnd can participate in 2 parallel signaling pathways within the loser cells: one that is Tak1 independent and reinforces the CSM, in which cell death is mediated by Rel (Meyer *et al.* 2014), and another that directs signaling through the canonical JNK pathway mediators and activates JNK reporters (de la Cova *et al.* 2004; Moreno and Basler 2004) but provides only a minor contribution to the death of loser cells. Our results make it clear that complex signaling inputs are deployed in the loser cells. This work thus helps to illuminate the qualitatively different survival and death cues that WT cells growing in the *tub > myc^STOP^>* context (i.e. loser cells) are subject to compared with genotypically identical WT cells in a noncompetitive environment.

## Discussion

Although the JNK pathway is activated during Myc cell competition, the relative contribution of this pathway to the elimination of the WT loser cells and how JNK signaling and the CSM interact have remained unclear. We have attempted to understand these interactions with a series of genetic experiments. We establish that loss of Egr/TNF or Grnd/TNFR suppresses the elimination of loser cells in the competitive *tub > myc^STOP^>* background, indicating that each factor is functionally required in the process. Although Egr is known to kill cells via JNK signaling, our experiments indicate that JNK pathway activity has only a minor role in the death of WT loser cells. Our results suggest that in Myc cell competition assays, signaling in the loser cells by Egr/Grnd bifurcates upstream of Tak1, to facilitate and/or potentiate the activity of the CSM in a process that is independent of downstream JNK signaling activity. Collectively, our experiments bolster earlier findings that abolishing JNK signaling is not sufficient to prevent the elimination of loser cells in this competitive context (de la Cova *et al.* 2004). Future studies will be important to determine the mechanistic basis of the interactions between Egr/Grnd and the CSM.

Our experiments indicate that Grnd functions as the primary Egr receptor in wing discs in this experimental context. Grnd is highly and ubiquitously expressed in wing discs, whereas the TNFR Wgn expression appears to be more restricted. The expression of *wgn* in the WP (Fig. 2a), where cells are most severely subject to Myc supercompetition (Alpar *et al.* 2018), raises the possibility that Wgn and Grnd could act semiredundantly in that region to promote loser cell elimination. However, the loss of Wgn is inconsequential in loser cells (Fig. 2b and c), whereas loss of Grnd prevents their competitive elimination (Fig. 2e and f). The TNF ligand, Egr, is expressed in multiple tissues in the larva and can be produced as a membrane-bound protein or in a soluble form after proteolytic processing (Kauppila *et al.* 2003; Narasimamurthy *et al.* 2009). Although Egr is highly expressed in the AMP cells that reside under the notum region of the wing disc (Fig. 1d and i), its expression is below the limit of our detection in most epithelial cells of the disc. We were also unable to detect any *egr* mRNA expression in either loser (or winner) cells in the competitive context (Fig. 1i and i'). This suggests that *egr* expression is not an autocrine product of CSM activation in the loser cells. Instead, we speculate that Egr, which appears to be constitutively present in the circulating hemolymph (Agrawal *et al.* 2016), is produced and secreted by another tissue into the circulating hemolymph that bathes all larval tissues. One possibility is that circulating Egr signals in discs via Grnd to induce the low, tonic JNK activity observed in the wing disc (Fig. 1g and h) (McEwen and Peifer 2005; Willsey *et al.* 2016). If so, perhaps it serves to "prime" the loser cells in a competitive environment to the effects of the CSM.

The possibility that Egr is secreted from a remote tissue has precedence in previous work, in which Egr, synthesized in the larval FB, can lead to signaling in cells of the brain or wing disc (Agrawal *et al.* 2016; de Vreede *et al.* 2022). Grnd is present on the apical membranes of wing disc cells, which face inwards toward the disc lumen, and is also found in intracellular vesicles (Palmerini *et al.* 2021). Recent work suggested the apical localization of Grnd prevents easy access to secreted Egr unless cell polarity is lost, as in *scribbled* mutant discs, in which case Grnd mislocalizes to the basolateral membrane (de Vreede *et al.* 2022). Such mislocalization does not occur in either the loser cells or in the Myc-expressing supercompetitor cells in the competitive

assay used here (see Fig. 2d'), suggesting there is a different mechanism for the association of Egr–Grnd in this context.

Regardless of its tissue source, our results suggest a mechanism for how Egr contributes to the death of loser cells in wing discs. We speculate that once Egr binds to and activates Grnd on wing disc cells, the information is transmitted within the cells in 2 ways: in the competitive environment, Egr/Grnd activity reinforces CSM-mediated death of the loser cells; at the same time, a low level of JNK activity is stimulated by Egr/Grnd (Fig. 1g and h) that could also sensitize loser cells to the CSM (de la Cova *et al.* 2004). How do winner cells escape the cell death induced by the CSM and JNK activity? We suggest 2 nonexclusive possibilities. First, given the propensity of increased Myc expression to suppress Egr-JNK signaling (Huang *et al.* 2017), cells carrying the >*myc*$^{STOP}$> cassette may be less susceptible to the death-inducing (or sensitizing) role of JNK activity. Second, although the >*myc*$^{STOP}$>-expressing winner cells provide the cues that trigger CSM activation, the CSM's killing activity is restricted to the loser cells (Meyer *et al.* 2014; Alpar *et al.* 2018), biasing any augmentation of its activity to those cells. Interestingly, the elimination of loser cells appears to be stochastic, with some loser cells falling to the death-inducing signals more easily than others. The probability of loser cell death is likely influenced by several independent factors, such as accumulation of the proapoptotic factor Hid, which would in turn be influenced by inputs to the *hid* locus via the activities of the CSM and JNK signaling.

Our findings provide another example of productive cross-talk between TNF/TNFR and NF-κB signaling pathways, which occurs in both the mammalian and *Drosophila* innate immune responses and, as in the cell competition assays, also diverge at the level of Tak1 and Tab2 (Myllymaki and Ramet 2013). One possibility is that downstream of Egr/Grnd activation, Traf4/6 facilitates and/or enhances CSM-mediated Dredd activity in the loser cells, thereby augmenting Rel-dependent, loser-specific gene expression and cell death. Intriguingly, although Dredd and Rel function in the the IMD immune response and in the CSM, neither Tak1 nor the IKK complex is required for loser elimination in Myc cell competition (Meyer *et al.* 2014), perhaps illustrating again how these pathways can be repurposed for different biological tasks.

We note that the intersection of the CSM and JNK signaling that we describe here may be unique to the context of Myc-induced competition. Competition between the polarity deficient *scrib* mutant cells and WT cells also utilizes Egr-JNK signaling to eliminate *scrib* loser cells, but this is Tak1-dependent and also requires Wgn (Igaki *et al.* 2006). Future studies will be important to sort out these differences. Nonetheless, our results help to explain the basis of the relative contributions of the CSM and JNK activity to Myc-induced cell competition. In addition, they add to the growing body of work that illustrates the remarkable signaling flexibility that occurs between cells in competitive contexts, allowing cells to tune their response to a given cellular or environmental context.

## Data availability

The resource origin and associated information are described in the reagents table (in Supplementary material). Fly strains are described in the article and are publicly available at the Bloomington Drosophila Stock Center, the Vienna Drosophila Resource Center, or upon request. Reagents used in this study are publicly available. Raw data and metadata from images are available from the corresponding author upon request.

Supplemental material is available at GENETICS online.

## Acknowledgments

We thank P. Leopold, B. Lemaitre, M. Miura, M. Milan, T. Igaki, and D. Bohmann for *Drosophila* strains and P. Leopold for anti-Grnd antibodies. We are grateful to members of the Johnston lab for helpful advice, C. Cary, P. Guevarra, D. Miranda, and J. Park for excellent technical support. We are indebted to the BDSC (supported by NIH P40OD018537), DGRC (supported by NIH 2P40OD010949), and FlyBase (Öztürk-Çolak *et al.* 2024) for their valuable resources.

## Funding

Funding for this study was provided by grants from the National Institutes of Health (R01GM078464 and R35GM131871) and the National Cancer Institute (R01CA192838) to LAJ.

## Conflicts of interest

The authors declare no conflicts of interest.

## Author contributions

All authors conceived and designed experiments. ALK ASS, CdlC, and MZ conducted experiments and analyzed the results. ALK ASS, CdlC, and LAJ contributed to writing the paper. LAJ supervised the project and provided grant support.

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

*Editor: B. Calvi*