## [Peer Review File · Genetics]

The *Drosophila* TNF Eiger promotes Myc super-competition independent of canonical JNK signaling

Albana Kodra, Aditi Sharma Singh, Claire de la Cova, Marcello Ziosi, and Laura Johnston

NOTE: The reviews and decision letters are unedited and appear as submitted by the reviewers.

In extremely rare instances and as determined by a Senior Editor or the EIC, portions of a review may be redacted. If a review is signed, the reviewer has agreed to no longer remain anonymous.

The review history appears in chronological order.

Review Timeline:

Submission Date:	2024-04-12
Editorial Decision:	2024-05-13
Resubmission Received:	2024-06-11
Editorial Decision:	2024-06-14
Revision Received:	2024-06-19
Accepted:	2024-06-21

May 13, 2024

GENETICS-2024-307025

The Drosophila TNF Eiger contributes to Myc super-competition independent of JNK activity

Dear Dr. Johnston:

Experts in the field have reviewed your manuscript and I have read it as well. I am pleased to inform you that, with minor revisions, it is potentially suitable for publication in GENETICS. The reviewers have comments and concerns that need to be addressed in a revised manuscript. You can read their reviews at the end of this email.

It is most important that you address the following in your resubmission:

- 1) The major issue that all three reviewers mentioned was if and how Grnd is connected to Dredd / CCSM in the context of super competition. They were not convinced by the evidence in Fig 5. In the absence of some experimental evidence for this connection, this gap in knowledge would need to be acknowledged and the interpretation softened.
- 2) Reviewer 3 thought it important to use RNAi to map the tissue source of Egr and validate the negative results from the TAK l.o.f experiments. I think that the latter is the most important critique to address.
- 3) The reviewers had a number of other questions and suggestions that you should address in some fashion, including issues with reference formatting. I also list a few things after the reviews that I hope you will find helpful during revisions.

We look forward to receiving your revised manuscript. Please let the editorial office know approximately how long you expect to need for revisions.

Upon resubmission, please include:

1. A clean version of your manuscript;
2. A marked version of your manuscript in which you highlight significant revisions carried out in response to the major points raised by the editor/reviewers (track changes is acceptable if preferred);
3. A detailed response to the editor's/reviewers' comments and to the concerns listed above. Please reference line numbers in this response to aid the editors.

Additionally, please ensure that your resubmission is formatted for GENETICS.

<https://academic.oup.com/genetics/pages/general-instructions>

Follow this link to submit the revised manuscript: Link Not Available

Sincerely,

Brian Calvi
Associate Editor
GENETICS

Approved by:
Meera Sundaram
Senior Editor
GENETICS

Reviewer #1 (Comments for the Authors (Required)):

In the article "The Drosophila TNF Eiger contributes to Myc super-competition independent of JNK activity," Kodra et al provide evidence for a novel signaling pathway - the Cell Competition Signaling Pathway (CCSM). This signaling pathway utilizes components of both the Toll and IMD/JNK pathways to induce cell death of "loser" cells during development when in competition with more fit myc-overexpressing "winner" cells.

This paper does a series of well-designed and executed studies using a variety of different reporters, labels, and genetic tools to reconfirm and establish for the first time the requirements for Eiger, Grindelwald, Traf4, and Traf6 in competition based cellular elimination. Cell death of the "losers" was not dependent on Wengen, the JNKKK Tak 1, JNKK Hemipterous, or JNK Basket.

The studies instead suggest that Dredd signaling plays a role in competition-induced apoptosis.

The most interesting and novel part of the paper is the suggestion JNK activation is not directly connected to competitive status and that instead cellular elimination relies on a combination of JNK and TLR signaling components. Unfortunately, the data in this paper only suggests this possibility rather than clearly demonstrating and elucidating the CCSM.

My main concern about claiming true JNK independence and a reliance instead on Dredd is the reliance on studies using overexpression of Dredd and an RNAi knockdown of Grindelwald Dredd overexpression studies. Figure 5.

- The authors assert that overexpression of Dredd enhances "loser" cell elimination. This is not obvious from the data and there appears to be no direct comparison of WT clone area compared to +Dredd clone areas. The reduced clone area of the +Dredd controls suggests an overall diminished area which would make determining an enhanced elimination more difficult. Statistical analysis across WT and +Dredd or between WT control v losers and +Dredd control v losers would be necessary to provide evidence for this.
- The lack of significance between the grndIR control clone area size and the losers (Figure 5F) is used to argue competition is Grnd-dependent. However, the lack of significance appears to rely on a few outliers in each category. There also appear to be a disproportionate number of experimental repeats in these categories, which leads to a questioning of how robust this lack of difference is.
- When combining overexpression of Dredd and an RNAi knockdown of Grindelwald, there appears to be some complicating factors at play. Combined overexpression and knockdown appear to lead to reduced control clone area compared to Dredd+alone control clone area. This is interesting given that GrndIR alone does not lead to any reduction in control clone area. While statistically this may not be the case, it would be important to run these tests if using the control as a standard against which to compare elimination.
- The statistical difference between control and loser clone size in Dredd+GrndIR also seems to have required many more tests to establish than any other of the comparisons. I am concerned that experiments were repeated until the statistics worked out rather than there still being robust elimination due to cellular competition.
- There is a lack of a Data Availability statement that would allow readers to dive deeper into this.

Even beyond the concerns with Figure 5, the limitations of this model should be considered. Super-competition is just one of the types of cellular competition being studied. There is significant evidence this competition and its signaling pathways may be distinct from Xrp1-dependent or other forms of cellular elimination. It should also be noted that studies have shown an interplay between Myc and JNK. The reliance of a system where even modest overexpression of Myc is what is used to drive winner status could influence JNK signaling pathways.

Without a clear link between Grindelwald/Traf6/Traf8 activation and Dredd signaling, the paper suggests but does not supply strong evidence for their proposed CCSM. The work is interesting and, with the exception of Figure 5, is good. However, their main assertion of a but not as convincing.

My other comments are more minor.

- A lack of a direct correlation between JNK transcriptional activity and "winner" vs "loser" status (especially given that there is some low-level activity in all cells) does not clearly eliminate the possibility that JNK activity is still central to cellular elimination due to competition. (Figure 1F-G).
- The importance and relevance of ectopic expression of *efr* in Figures 4B-C is unclear. This introduces a new approach whose direct comparison and relationship to the other studies looking at clonal populations in the wind disk is unclear.
- There is no mention of the *grndIR* flies in the methods.
- It would be helpful to explicitly state that a constitutively active form of the intracellular domain of Grnd is being used rather than just an intracellular domain.
- Elimination is misspelled at the bottom of page 9.

Reviewer #2 (Comments for the Authors (Required)):

Critique of Kodra et al

Kodra and colleagues present new and important finding of the mechanism of Myc super-competition. Super-competition is the process by which cells with higher fitness due to increases in genetic dose of Myc eliminate wild-type (WT) neighboring cells. The process of cell competition and super-competition play important roles in a wide array of processes (from embryonic development to tumor progression) but there are critical gaps in our knowledge about the molecules involved. Previously the Johnston lab has identified what they term the cell competition signaling module (CCSM) comprising some components of Toll and IMD pathways in the elimination of losers. These are: several Toll related receptors (TRRs), the secreted Toll ligand Spatzle (Spz), and 2 spz-cleaving proteases, the adaptor FADD, the caspase-8 homolog Dredd, and the NF- κ B factor, Relish (Rel). The role of the JNK pathway in Myc super-competition is not well understood, and the authors set out to test its function using a set of elegant genetic experiments. The authors find (1) the TNF-like factor Eiger (Egr) is required in WT losers for their out-competition; (2) The Egr receptor Grnd is also required in WT losers for their elimination but the other TNFR in flies - Wgn - is not; (3) expression of the Grnd intracellular domain is sufficient to induce death in non-competitive and competitive scenarios; (4) Traf4 and Traf6 are required in losers for their elimination; (5) surprisingly, the Tak1 kinase that acts downstream of

Egr/Grnd does not regulate loser death in competitive scenarios, and reduction of JNKK (Bsk) and JNK (Hep) cannot suppress the death of losers. These data suggest that Egr/Grnd function in losers is independent of JNK activity, and indeed the authors provide data to support this model in the last figure of the paper. Taken together these results support a model in which activation of Grnd by Egr during cell competition leads to activation of the FADD/Dredd/Rel and then apoptosis (hid) and in parallel to the activation of JNK signaling, which modestly contributes to loser death. Overall, the data are of high quality, the assays are robust, and the genetics are very solid. These results advance the field of cell competition and would be of interest to a wide range of geneticists.

I have some suggestions for the authors.

1. Some of the references are properly formatted but others lack the publication year or still have the brackets of the bibliographic software (Endnote).
2. I suggest that the authors move the model from Suppl Fig 1 to main body Fig. 1. I also suggest that they add the Spz proteases, etc so that it matched the text. When the authors write about Egr/Grnd activating the CCSM, it is not clear to me whether this proceeds from Egr/Grnd to Spz/TI to FADD/Dredd/Rel/Hid or whether Spz/TI is not involved. In other words, I am asking for a bit more precision about how Egr/Grnd activates CCSM. I think at some clarity in the model would be helpful to the readers.
3. The figures are pixelated so I suggest uploaded higher resolution images.
4. The authors need to provide statistically analysis of Fig. 5D vs. Fig. 5B since they use the word "significantly" but do not provide a P value.

Reviewer #3 (Comments for the Authors (Required)):

The manuscript of Laura Johnston and colleagues addresses the role of the TNF/JNK signalling pathway in dMyc driven cell competition using the wing primordium of *Drosophila* as model system. Authors conclude that Eiger (not coming from the wing epithelium, although this should be tested by RNAi), its receptor Grindelwald (but not Wengen), Traf4 and Traf6 (but not Hep and JNK) are required in loser cells to undergrow and get outcompeted. Paper is well written, figures self-explanatory and this topic is timely for the Genetics journal.

I have the following comments, though, that should be raised by the authors

- (1) Intro is, to a large extent, devoted to summarize previous results from the field including the publications of their own lab on the CCSM. I would suggest to make the Intro shorter and to the point of the potential role of Eiger/JNK axis in cell competition. Otherwise, the reader might get unfocused.
- (2) The proposal that Eiger is not functioning in a paracrine manner should be reinforced by functional data (RNAi for eiger?)
- (3) The no-role of Tak1 and Tak2 should be reinforced by the use of RNAi forms of these two genes. The fact that mutant larvae are viable either indicates that mutants are not completely null or there is strong maternal contribution.
- (4) I am not fully convinced about the conclusion that Hep and JNK have no role. There is a clear partial rescue of the size of loser clones!! Again, the fact that the hep allele is viable indicates that this mutant is not completely null or there is strong maternal contribution. The use of RNAi forms of hep and bsk should be able to reinforce the data
- (5) I am no convinced by the data on the potential connection Dredd-Grindelwald present in Figure 5. I think it should be removed.
- (6) References are not ok.

Associate Editor Comments:

I think that the manuscript could be improved by editing for clarity in some places. Here are a few suggestions...

pg 3

"Reporters of the JNK stress pathway such as puc-lacZ, an enhancer trap insertion in the puckered (puc) locus (encoding a phosphatase antagonist of JNK activity) (MARTINBLANCO et al. 1998; MCEWEN AND PEIFER 2005) indicate that JNK activity is present in some epithelial cells of the wing and eye imaginal discs during their normal growth, where it promotes cell survival by blocking JNK activity (MCEWEN AND PEIFER 2005; WILLSEY et al.)."

I am guessing that the "it" in the last clauses refers to puc in the first clause.

pg 9

"However, in the egr3AG background, the egr mutant clones generated in both competitive and non-competitive contexts grew at the same rate and to a similar final size"

I think that calling these egr mutant clones is a bit confusing. They are egr mutant, but so is the rest of the tissue.

pg 13

"If so, we reasoned that in this competitive context, activation of Egr/Grnd signaling in a Tak1 mutant should not suppress loser cell death (i.e., the cells would still be eliminated)."

I suggest rewording to "...a Tak1 mutant should not suppress Egr/Grnd induced cell death."

Fig 1: The order of the legend does not match figure panels.

Fig 3E,F - I find it a little confusing that the insets to these panels are different discs and time points. I suggest making them separate panels with time after induction noted.

Fig 4I-J - There is a legend for these panels, but no panels in the figure.

Include a data availability statement.

Please update FlyBase ref to: <https://academic.oup.com/genetics/article/227/1/iyad211/7596147?login=true>

FYI: Genetics has switched to lower case letters for figure panels.

Kodra et al, authors rebuttal

The authors thank each reviewer and the Associate Editor for their comments and insightful recommendations. Below we first describe the overall changes that we made to the manuscript, and then we provide answers (in black text) to specific comments from each reviewer (in blue italic text).

Overall changes to the manuscript:

1. Reviewers 1 and 3 expressed issues with the *UAS-Dredd + UAS-grnd^{IR}* experiments, and Reviewer 3 suggested removing the experiment from the paper. We agree with both reviewers that the data are not without problems that can be solved within an appropriate timeframe, so we have taken their advice and removed this experiment from the paper (this removal does not alter the major conclusions of the work). We have therefore re-organized the figures.
2. We have changed all graphical measurements to microns for uniformity.
3. We have reformatted and/or rearranged figures for clarity, and formatted figure panel labels according to the *Genetics* format.
4. We have updated and clarified the fly strains list, and corrected typos.
5. In Fig. 3 we have added new panels to replace the inserts, as suggested by the editor.
6. We have updated the FlyBase reference as requested by the editor.
7. We have added a data availability statement.

Reviewer 1.

1. The authors assert that OE Dredd enhances loser cell elimination.... Statistical analysis across WT and +Dredd or between WT control v losers and +Dredd control v losers would be necessary to provide evidence for this.

This experiment is now removed from the paper, as noted above.

2. The lack of significance between the grndIR control clone area size and the losers (Figure 5F) is used to argue competition is Grnd-dependent. However, the lack of significance appears to rely on a few outliers in each category. There also appear to be a disproportionate number of experimental repeats in these categories, which leads to a questioning of how robust this lack of difference is.

We have moved the *UAS-grndIR* data to Fig. 2f, putting it more appropriately alongside the *Grnd Df/Mi* data in Fig. 2e. We appreciate the reviewer's comments about the repeats and outliers. We try to analyze approximately the same number for each experiment. For this experiment, the same number of clones were analyzed for each of the genotypes (28 *grndIR* Ctls and 28 *grndIR* losers).

As for the outliers, they are a natural outcome of clonal analysis and are most often due to fusion of clones during the growth period. Because clonal growth is subject to many variables that can affect clone size and number (food quality, growth temperature, larval crowding, duration of the initiating heat shock, as well as precise developmental timing), we have optimized to the best of our ability the conditions for generating clones (as detailed in the Methods) and try hard to account for all relevant parameters when we interpret the results of our experiments. One way to assess the data is with frequency distribution histograms of clone sizes – which illustrate that outliers are commonly observed for each genotype shown here. This is the primary reason why we measure the **median** clone size rather than the **mean** (see Neufeld et al 1998, de la Cova et al 2004). It is also the reason we carry out non-parametric tests for significance. For example, the WT loser distribution is sharply skewed to the left, due to the increased number of small clones compared to WT ctls (part of the loser phenotype). By contrast, the size distributions of *grnd^{IR}* ctls and *grnd^{IR}* losers are very similar, and also similar to the WT ctls – nicely showing suppression of the loser phenotype. These relationships between control and loser clones - or any cell clones exhibiting proliferation, growth or survival differences - are typical for experiments using clone surface area measurements. To address the point of the robustness of the result with *grnd^{IR}*, we point out that cell competition is also robustly suppressed in the Grnd mutants (Fig. 2e).

3. When combining overexpression of Dredd and an RNAi knockdown of Grindelwald, there appears to be some complicating factors at play. Combined overexpression and knockdown appear to lead to reduced control clone area compared to Dredd+ alone control clone area. This is interesting given that Grnd^{IR} alone does not lead to any reduction in control clone area. While statistically this may not be the case, it would be important to run these tests if using the control as a standard against which to compare elimination.

Upon the advice of the reviewers, this dataset was eliminated from the paper (see note above).

4. Even beyond the concerns with Figure 5, the limitations of this model should be considered. Super-competition is just one of the types of cellular competition being studied. There is significant evidence this competition and its signaling pathways may be distinct from Xrp1-dependent or other forms of cellular elimination. It should also be noted that studies have shown an interplay between Myc and JNK. The reliance of a system where even modest overexpression of Myc is what is used to drive winner status could influence JNK signaling pathways.

We completely agree with the reviewer that the signaling module we call the CSM and its interactions with JNK, Egr and Grnd might be specific to Myc super-competition. It is very clear that many aspects of Myc super-competition are different than competition between WT cells and *Rp/+* cells, for several reasons (including, but not limited to that the former has no developmental delay, no requirement for Xrp1 (unpublished data), and a modest requirement for JNK activity, as we first reported in *de la Cova et al 2004* and now elaborate more here. In the Discussion (page 17 of the ms.), we tried to make that point very clear, by stating that this relationship between the CSM and JNK may well be unique to Myc super-competition. On the other hand, JNK activation plays a prominent role in *Rp/+* cell competition, and since Meyer et al (2014) found that *Rp/+* cell competition also requires some innate immune components (dorsal/NFkB, cactus, Toll related receptors), it is entirely possible that interactions between those and components of the JNK signaling pathway could occur; we have not examined this in any detail but it would be interesting to explore this in the future.

5. A lack of a direct correlation between JNK transcriptional activity and "winner" vs "loser" status (especially given that there is some low-level activity in all cells) does not clearly eliminate the possibility that JNK activity is still central to cellular elimination due to competition. (Figure 1F-G).

We agree with the reviewer, and have used conditional language that we hope makes this clear in the text. However, in this competitive context, the genetic experiments argue against a major role for JNK in the loser death: significant loser elimination continues even in mutants where it is clear that most JNK activity is abolished.

6. The importance and relevance of ectopic expression of *egr* in Figures 4B-C is unclear. This introduces a new approach whose direct comparison and relationship to the other studies looking at clonal populations in the wing disk is unclear.

We have elected to keep this figure in the paper because it illustrates both the dramatic effect that Egr expression can have in the wing disc, and the very strong suppression of those effects in the *hep^{r75}* mutant.

Reviewer 2.

1. *References need formatting.*

Thank you for pointing this out, we have reformatted them. Unfortunately the Endnote program (version X9) that we use refuses to imbed the dates when references are cited in the text; however they are correctly cited in the References section. We hope that the copy editors can help us solve the problem.

2. *Add Supp fig to main Fig. 1, also add SPE and ModSP.*

We have added the model schematic to Figure 1 and also added SPE and ModSP, as requested.

3. *figures are pixelated.*

We apologize, the images submitted were pdf versions and we admit did not come across as hoped. They have now all been formatted in Photoshop and Adobe Illustrator, with images at 350dpi and the model schematic at 600 dpi as per *Genetics* formatting.

4. *statistical analysis of Fig5D and B.*

Please note that these data are now shown in Fig. 4; Fig. 5 has been deleted. We have added the P values for this experiment (now **Fig. 4j-l**) to the figure legend. Mann-Whitney tests between WT ctls vs *Tak1² + grnd^{intra}* ctls indicate they are not significantly different ($p = 0.3256$).

Reviewer 3.

1. *Make the Intro shorter and to the point of the potential role of Eiger/JNK.*

We agree with the reviewer and have revised our Introduction accordingly (p. 2).

2. *The proposal that Eiger is not functioning in a paracrine manner should be reinforced by functional data.*

We have considerable data regarding this issue and plan to submit a manuscript soon that describes experiments that determine the tissue source of Egr (noted in the text as Sharma Singh et al, ms in prep). We initially thought about including that data in this ms., but the experiments have turned out to be quite complex and numerous, thus we feel that an exploration of Egr's paracrine function in cell competition is better suited for a separate paper. In addition, the intent of the current work is to focus on the genetic intercalations between then CSM and JNK signaling downstream of Egr/Grnd.

3. The no-role of Tak1 and Tak2 should be reinforced by the use of RNAi forms of these two genes. The fact that mutant larvae are viable either indicates that mutants are not completely null or there is strong maternal contribution.

With respect to the reviewer, we do not think that this experiment is necessary. Our data here demonstrate that the *Tak1*² mutant suppresses the effect of expression of Grnd^{intra} in clones (Fig. 4I). Our data are in complete agreement with Andersen et al 2015, who used *Tak1* RNAi to show, similarly, that loss of Tak1 suppresses the effect of expression of Grnd^{intra}. To provide information about the mutants: the two different *Tak1* alleles that we used in this work (*Tak1*¹ and *Tak1*²) carry mutations that inactivate the Tak1 kinase domain, and both behave as phenotypic nulls (Vidal 2001). They are each well-characterized as inhibitors of JNK activation (Vidal 2001, Silverman 2003, Delaney 2006, Stronach 2014).

4. I am not fully convinced about the conclusion that Hep and JNK have no role. There is a clear partial rescue of the size of loser clones!! hep^{R75} also viable, so probably not full null, need RNAi to validate.

We agree with the reviewer that there does seem to be a partial rescue of the loser clone size, and have made that more clear in the text.

hep^{R75} is a P element excision allele, well characterized as a strong loss of function, and frequently used to eliminate Hep function (<http://flybase.org/reports/FBal0045852.htm>) in the literature. In addition, RNAi can often be less effective than strong LOF mutants.

5. I am not convinced by the data on the potential connection Dredd-Grindelwald presented in Figure 5. I think it should be removed.

Done as requested.

6. References are not OK.

We have reformatted the references, but please see the note above.

June 14, 2024

RE: GENETICS-2024-307179

Dear Laura:

I am pleased to accept your manuscript entitled "The Drosophila TNF Eiger promotes Myc super-competition independent of canonical JNK signaling" for publication in GENETICS, pending minor revision. I think that you did a great job revising the manuscript in response to reviewers concerns.

One of the last things to fix are the figure 2 panel labels which need to be reordered to match text and figure legend (i.e. b, b' vs c, c', it happens).

One minor suggestion is that you could be even more explicit on first definition of "winner" and "loser" clones so readers remember. The fact that the genotype of winner cells are 3x Myc and losers are 2x Myc is only mentioned once after that in both text and figure legends. You may also wish to remind the reader in select places. I also wouldn't choose to call those transformed flies wild type (WT), although I understand that you mean not mutant for other genes.

I am sorry to hear about your continued Endnote problems. I will leave it to the editorial office to advise how to proceed. Hopefully copy editors can help to fix the references in text.

Follow this link to submit the revised manuscript: Link Not Available

Thank you for submitting this story to Genetics.

Sincerely,

Brian Calvi
Associate Editor
GENETICS

Approved by:
Meera Sundaram
Senior Editor
GENETICS

Reviewer comments:

Associate Editor comments:

June 18, 2024

RE: GENETICS-2024-307179

Dear Brian,

Many thanks for your acceptance of our paper! We have fixed the issues you raised for Fig. 2, and also have mentioned the 3xMyc vs 2xMyc in a few places in the manuscript to help the reader. I am uploading a marked and a clean copy of the final ms. and also a corrected version of Fig. 2.

Thanks again to you and Genetics for the speedy review process! We look forward to seeing in in print.

Best wishes,
Laura

June 21, 2024

RE: GENETICS-2024-307179R1

Prof. Laura A. Johnston
Columbia University Medical Center
Genetics & Development
701 West 168th Street, HHSC 704
New York, New York 10032

Dear Dr. Johnston:

Congratulations! We are delighted to inform you that your manuscript entitled "The Drosophila TNF Eiger promotes Myc super-competition independent of canonical JNK signaling" is acceptable for publication in GENETICS. Many thanks for submitting your research to the journal.

To Proceed to Production:

1. Format your article according to GENETICS style, as discussed at <https://academic.oup.com/genetics/pages/general-instructions>, and upload your final files at <https://genetics.msubmit.net>.
2. Your manuscript will be published as-is (unedited-as submitted, reviewed, and accepted) at the GENETICS website as an Advanced Access article and deposited into PubMed shortly after receipt of source files and the completed license to publish. Please notify sourcefiles@thegsajournals.org if you do not wish to publish your article via Advanced Access.
3. We invite you to submit an original color figure related to your paper for consideration as cover art. Please email your submission to the editorial office or upload it with your final files. You can submit a small-sized image for evaluation, and if selected, the final image must be a TIFF file 2513px wide by 3263px high (8.375 by 10.875 inches; resolution of 600ppi). Please avoid graphs and small type.

If you have any questions or encounter any problems while uploading your accepted manuscript files, please email the editorial office at sourcefiles@thegsajournals.org.

Sincerely,

Brian Calvi
Associate Editor
GENETICS

Approved by:
Meera Sundaram
Senior Editor
GENETICS

note: Please add jnls.author.support@oup.com and genetics.oup@kwglobal.com (or the domains @oup.com and @kwglobal.com) to your email program's "safe senders" list. You will be contacted by both at various points during the production process.

Review comments (if applicable):